# Computational Study of Aerodynamic Effects of the Dihedral and Angle of Attack of Biomimetic Grids Installed on a Mini UAV

**DOI:** 10.3390/biomimetics9010012

**Published:** 2023-12-29

**Authors:** Rafael Bardera, Ángel Antonio Rodríguez-Sevillano, Estela Barroso Barderas, Juan Carlos Matias Garcia

**Affiliations:** 1Instituto Nacional de Técnica Aeroespacial (INTA), Experimental Aerodynamics, Torrejón de Ardoz, 28850 Madrid, Spain; barrosobe@inta.es (E.B.B.); matiasgjc@inta.es (J.C.M.G.); 2Escuela Técnica Superior de Ingeniería Aeronáutica y del Espacio (ETSIAE), Universidad Politécnica de Madrid (UPM), 28040 Madrid, Spain; angel.rodriguez.sevillano@upm.es

**Keywords:** UAV, biomimetic, wingtip, grids, dihedral, angle of attack, aerodynamic efficiency, CFD

## Abstract

In this paper, a numerical analysis of a biomimetic unmanned aerial vehicle (UAV) is presented. Its wings feature three grids at the tip similar to the primary feathers of birds in order to modify the lift distribution over the wing and help in reducing the induced drag. Numerical analysis using computational fluid dynamics (CFD) is presented to analyze the aerodynamic effects of the changes in dihedral and angle of attack (with respect of the rest of the wing) of these small grids at the tip. The aerodynamic performances (lift, drag, and efficiency) and rolling capabilities are obtained under different flight conditions. The effects of changing the dihedral are small. However, the change in the grid angle of attack increases aerodynamic efficiency by up to 2.5 times when the UAV is under cruise flight conditions. Changes to the angle of attack of the grids also provide increased capabilities for rolling. Finally, boundary values of the pressure coefficient and non-dimensional velocity contours are presented on the surfaces of the UAV, in order to relate the aerodynamic results to the aerodynamic patterns observed over the wing.

## 1. Introduction

The evolution of unmanned aerial vehicles (UAVs) has become essential in the global aerospace landscape. Initially developed exclusively for military applications, recent technological advances have facilitated their integration into commercial activities [1]. UAVs now serve as indispensable tools for executing tasks in hazardous, monotonous, or otherwise challenging environments, mitigating human risks [2]. The aerodynamics of small UAVs, owing to their unique size and operational velocities, are characterized by low Reynolds numbers, necessitating innovative design solutions [3,4] that strike a balance between low Reynolds numbers and low aspect ratios [5].

In recent years, the field has witnessed a surge in bio-inspired aircraft that emulate the flight mechanics of animals; these have emerged as a captivating avenue within aviation technologies [6,7]. These bio-inspired developments, specifically applied to UAVs, constitute a critical facet in the ongoing pursuit to enhance vehicle performances by elevating aerodynamic efficiency while concurrently reducing weight, emissions, and operational costs. Analogous to traditional aircraft, a considerable body of research is dedicated to minimizing the aerodynamic drag of UAVs. As an example, the adaptability of birds in various flight conditions is analysed in [8]. The authors reconstructed an eagle wing’s inner portion during a rapid pitch-up maneuver using photogrammetric techniques. The resulting mathematical model of the wing’s surfaces incorporated spanwise twists, bending, chord distribution variations, and aerofoil shapes. The bird aerofoil displayed a consistent drag coefficient ($C_{D}$) across a broad lift coefficient ($C_{L}$) range, highlighting its robustness. Using wind tunnel testing, Marco et al. [9] investigated the slotted wing tips of a jackdaw, commonly linked to improved soaring. The study confirmed individual wakes from separated primary feathers, indicating a multi-slotted function in both gliding and flapping flight. The resulting multi-cored wingtip vortex suggests enhanced aerodynamic efficiency. Considering benefits specific to flapping flight and the widespread presence of slotted wing configurations across species, the study proposed the hypothesis that slotted wings initially evolved to enhance performance in powered flight.

Taking inspiration from nature, where birds leverage slotted feathers to reduce drag force, researchers have adapted this concept into a novel wing grid solution [10,11]. By augmenting wingspan through the use of grids, studies have demonstrated a noteworthy reduction in induced drag, particularly during low angles of attack and cruise phase flight conditions [12,13,14]. As estimated in [15], induced drag constitutes around 30% of total drag during cruise flights and even more at lower speeds. Decreasing this aerodynamic factor would significantly reduce the operating costs of an aircraft. In the same work, the authors focused on increasing the effective aspect ratio of a wing, allowing it to have a lower wing area and weight in addition to higher cruise lift-to-drag ratios. An example of an optimization process for wingtip devices can be found in [16]. That study investigates the design of wing tip devices for high and low speeds using the fast aerodynamic prediction tool LIDCA (lift and drag component analysis). For this purpose, an airfoil database module based on two-dimensional flow simulations was integrated into an existing lifting-line method by means of multi-dimensional interpolation. The accuracy of the method was validated by detailed studies at high lifts and speeds. The most efficient wing tip designs were assessed across different flight conditions. Siddiqui et al. [17] extensively explored the use of wing tip devices inspired by birds and historical aeronautical advancements, aiming to boost aircraft performance by reducing induced drag. Their work focused on biomimetic and conventional methods, consolidating past analyses and experiments that assessed their impact on overall aerodynamic performance. Additionally, they outlined industry needs and past achievements and identified unexplored areas for future exploration, emphasizing the potential for innovation in advancing aviation technology.

Another study is the numerical work conducted by Reddy et al. [18], which was focused on maximizing the lift coefficient and the lift-to-drag ratio as well as minimizing the drag and moment coefficient by using split winglet designs featuring scimitar tip spikes. The winglet designs were obtained by varying the cant angle, leading and trailing edge sweeps, and length of scimitar spikes. The results indicated that a split winglet design disperses the vortex core more efficiently than a basic blended (horns up) winglet. When scimitar tip spikes are added to the split winglet, the wing tip vortex core radius increases, leading to improved flow redirection and lower induced drag. This suggests the potential for optimizing split winglets by incorporating multiple elements, resembling the spread feathers found at a soaring bird’s wingtips. Experimental studies based on the use of multi-winglets were conducted by Cosin et al. [19,20,21]. A baseline and six other different multi-winglets configurations were tested. The wingtip device was a variable configuration of multi-winglets with three tip sails; they allowed the adjustment of the cant angle and incidence for each sail independently. The winglet device led to improvements in the lift and a reduction in induced drag. The results led to a 32% improvement in the Oswald efficiency factor, signifying a notable increase of 7% in maximum aerodynamic efficiency. In addition, advancements of 12% in the maximum rate of climb and 7% in the maximum range were achieved.

Building upon these principles, a collaboration between the National Institute for Aerospace Technology (INTA) and the Technical University of Madrid (UPM) has culminated in the development of a biomimetic UAV specifically inspired by the primary feathers of birds [22,23,24,25,26], as shown in Figure 1. Figure 1 also illustrates how the grids can contribute to reduce induced drag, by decreasing the tip vortex size and enhancing overall aerodynamic efficiency.

Characterized by a rectangular wing featuring three semi-wings (grids), the UAV can modify its wingspan during flight, optimizing aerodynamic performance. By employing numerical comparisons facilitated by the Tornado software, previous research has determined the optimal chord, gap, and number of grids [23,26]. Figure 2 displays the UAV dimensions, showing a length of 275 mm and a variable wingspan of 540 mm (grids retracted) to 720 mm (grids fully extended).

The wing profile chord measures 90 mm [5], and the grids chord is 20 mm with 4.5 mm gaps between them. Moreover, the versatility of the UAV extends beyond drag reduction, and investigations have been conducted into its rolling capabilities when deploying grids selectively on one wing, as depicted in Figure 1 [24,25,27]. To complement the rectangular wing, the UAV incorporates a V-tail vertical stabilizer for longitudinal and lateral control.

In a previous study, variations in the grid span of each grid were analyzed, showing a 15% to 20% increase in the aerodynamic efficiency of the UAV during cruise and climb phases of the flight [28].

This paper takes a new step in UAV aerodynamic design optimization, by determining the effect of (1) a change of dihedral angle of each grid ($\delta_{1},\delta_{2},\delta_{3}$) and (2) the angle with respect to the wing of each grid ($\alpha_{1},\alpha_{2},\alpha_{3})$ (ordered from the grid placed nearest to the leading edge of the wing to the grid placed next to the trailing edge of the wing) on UAV performance. Both angles are defined in Figure 3. It uses computational fluid dynamics (CFD) to compute the lift, drag, and aerodynamic efficiency of the non-extended grids (base) and different configurations (by rotating the grids around the x-axis for dihedral and the y-axis for the angle with respect to the wing). Specifically, the configurations compared are presented in Figure 4 as base and standard grid configurations and the new modifications (tested with different values of the dihedral ($\delta =5^{\circ },10^{\circ },15^{\circ }$) and angle of attack of the grids with respect to the main wing ($\alpha =5^{\circ },10^{\circ },15^{\circ }$)). Finally, the grid configuration that achieves the maximum rolling coefficient and a flow visualization above the grids are also presented.

## 2. Numerical Set-Up

The computational analysis was executed utilizing the ANSYS-Fluent software. By exploiting a symmetry condition, only half of the model was computed due to the freestream velocity consistently aligning with the chordwise direction. The dimensions of the computational domain and surfaces are illustrated in Figure 5. To expedite convergence solutions and minimize memory, storage, and processing demands, the influence of the propeller was omitted.

Figure 5 also shows the utilization of an unstructured mesh, comprising approximately 5 million elements. A convergence analysis of the mesh independence was performed for the CFD model. In general, the same aerodynamic coefficients were obtained for grids with a number of elements greater than 4 million. Low-resolution meshes give poor results that overestimate the drag coefficient, but more defined meshes of 4, 5, and 6.5 million elements provide very similar values for the coefficients of the UAV forces, with minimal differences of 0.5%.

The chosen boundary conditions incorporated a free-stream velocity of $U_{\infty }=16$ m/s, resulting in a Reynolds number of $9.2\cdot 10^{4}$, characteristic of the normal operating regime for these types of vehicles. Near the aircraft body, sizing restrictions were imposed, dictating a maximum element size of 2 mm. For the boundary layer simulation, standard wall functions were implemented. The calculations employed a two-equation shear stress transport (SST) k − ω turbulence model [29]; this is one of the most commonly used turbulence models for the RANS equations. It uses a k − ω formulation in the boundary layer and k − ε for the free-stream, thereby avoiding problems at the inlet with free-stream turbulence properties. In addition, it exhibits good behavior in adverse pressure gradients and separating flow regions [30]. Thanks to the mesh refinement performed, the value of the wall parameter in the viscous sublayer was $y^{+}\leq 1$, where the k−ω standard model was used; $30<y^{+}<300$ was used in the logarithmic region, where the model used k−ϵ. Calculation settings included a coupled scheme for pressure–velocity coupling, and least-squares-cell-based spatial discretization. The numerical schemes used were second order upwind for the pressure, momentum, and turbulent kinetic energy. Finally, the solution converged in less than 500 iterations for each case computed, with a few hours of computation time.

The angle of attack (AoA) was modified by rotating the UAV model with respect to the incident free-stream velocity ($U_{\infty }$) using the same computational domain. In total, 36 simulations were performed under different wind conditions and configurations of the grid dihedral and angle with respect to the wing.

## 3. Results and Discussion

### 3.1. Base and Wingrid Configurations

In this section, the results for the base UAV configuration (no grids) and the grids in the standard configuration (dihedral and angle with respect to wing = 0) are presented and compared. The lift and drag aerodynamic coefficients ($C_{L}$, $C_{D}$) and aerodynamic efficiency ($E=C_{L}/C_{D}$) for both configurations are presented in Figure 6. To compare the results of the lift and drag coefficients regardless of the increase of wing surface when using the grids, the same wing surface value (S) for the UAV with grids was used for all cases tested.

The lift coefficient follows a linear trend between a −10° and 10° angle of attack (AoA), reaching near-zero lift at 0°. At 10°, the lift coefficient peaks, measuring 0.84 for the base configuration and 1.01 for the grid configuration. When the UAV is in descent flight with a negative AoA, the configuration with grids generates a higher negative lift; under a positive AoA, $C_{L}$ consistently surpasses its value without grids.

The minimum drag coefficient is obtained with a zero angle of attack for the base configuration, with a value of $C_{D}=0.051$; it is obtained with a higher value of $C_{D}=0.060$ for fully extended grids. The drag increases with positive and negative values of the angle of attack. This increase is slow during cruise flight conditions at low angles of attack ($0^{\circ }\leq \mathrm{AOA}\leq 5^{\circ })$ in all configurations; however, it gradually becomes more pronounced with higher and lower angles of attack.

Figure 6 also presents the aerodynamic efficiency (E) calculated as the ratio between the lift coefficient $C_{L}$ and drag coefficient $C_{D}$ under different AoAs. The goal of a proper UAV design is to generate as much lift as possible and reduce the drag. Looking into the results for aerodynamic efficiency, it remains similar for both configurations at angles less than −5° and greater than 10°. However, in the case of cruising AoA = $5^{\circ }$, the grid configuration manages to increase efficiency by 11% to a maximum of E = 7.8 (compared to 7.0 for the base configuration). Since the improvement in flight efficiency using the grids occurs in this small range of angles of attack, the simulations with different dihedral and grid angle configurations with respect to the wing will be tested in the range of AoAs between −5° and 10°, as marked on the aerodynamic efficiency plot in Figure 6. This is also the typical operating range of a UAV.

### 3.2. Dihedral Effect of Grids

This section analyses the effect of introducing dihedrals in the grids, with a progressive configuration from the forward grid to the aft grid of 5, 10, and 15°, respectively. Figure 7 presents the aerodynamic performances of the UAV base, grids, and dihedral grids configurations. In general, the lift coefficient under dihedral conditions shows very similar values to the case with grids. Only slightly different values are observed for ${-5}^{\circ }$ and $5^{\circ }$ AoAs. Aerodynamic drag increases with the dihedral configuration along the range studied. Finally, the aerodynamic efficiency with the dihedral is always between the minimum values of the base configuration and the maximum value of the configuration with grids. Numerically, with respect to the base configuration without grids, the grids with dihedral increase the efficiency by 0.2 but do not exceed the efficiency values of conventional grids without dihedral. Thus, it can be concluded that introducing a dihedral angle to the grids does not generate an advantageous aerodynamic effect upon the UAV studied.

### 3.3. Angle of Attack of the Grids

This section and the results contained in Figure 8 compare the configuration of grids with no dihedral ($\delta =0^{\circ }$) and no angle with respect to the wing ($\alpha =0^{\circ }$) against the four tested configurations of grid angles with respect to the UAV wing (GA5, GA10, GA15, and GA 5-10-15).

Firstly, the lift coefficient is higher for all cases where the angle of the grids (α) is increased. The effect is greater the smaller the AoA. In other words, the grid angle increases the $C_{L}$ in a significant way when the UAV is flying at $-5^{\circ }$ and $0^{\circ }$, but with less effect when the UAV is in ascent flight at $5^{\circ }$ or $10^{\circ }$. In general, the lift coefficient increases with the angle of the grids. This means that the greatest increase in lift is achieved with the maximum angles of GA 15 and GA 5-10-15, followed by GA 10 and GA 5. Similar to lift, the drag coefficient increases in all cases of grid angles tested. Moreover, it follows the same pattern: drag increases as the angle of the grids (α) increases. Furthermore, it becomes closer to zero in the ${-5}^{\circ }$ descent cases, allowing descent flight at low angles with a higher glide ratio.

Regarding $C_{L}$ and $C_{D}$, Figure 8 shows the aerodynamic efficiency. The case of cruising around 0° is particularly important because in all cases analysed there is a significant increase in aerodynamic efficiency. In general, the increase is again greater for the cases with a higher angle of the grids. Numerically, the GA5 configuration increases the efficiency by 1.4 points, the GA10 and GA15 cases increase it by 2.1 points, and the largest increase is experienced with progressive GA 5-10-15, with a 2.6 point increase in efficiency. However, when the UAV ascends (AoA = 5°), changing the angle of the grids no longer brings benefits, as they result in a lower aerodynamic efficiency than the base case. In this case, the higher the angle of the grids, the lower the aerodynamic efficiency. The same effect can be seen at higher angles of attack (AoA = 10°), where an increase in grid angle has no aerodynamic benefit.

In summary, increasing the angle of attack of the grids relative to the wing has very significant benefits in cruise flight, increasing the aerodynamic efficiency; this can be translated into a longer range, endurance, and lower energy consumption. In addition, during descent flight, the efficiency can be brought closer to 0, making the descent more controlled, without generating abrupt negative lift. In short, it provides greater controllability during UAV operations.

### 3.4. Rolling Capabilities Using Grids

In addition to improving aerodynamic performance, the grid angle can be changed on only one side of the wing to achieve increased lift and provide rolling moment to the UAV. In this way, the roll coefficient ($C_{l}$) generated by the grids is obtained from the simulations using
(1)$$C_{l}=\frac{L_{roll}}{\frac{1}{2}\rho V^{2}Sb}=\frac{L_{g}\cdot d}{\frac{1}{2}\rho V^{2}Sb}$$
where $L_{roll}$ is the rolling coefficient obtained from the lift generated by the grids $L_{g}$, the distance from the center to the grids is d=315 mm, ρ=1.225 kg/m^3^ is the air density, V=16 m/s is the flight velocity, S=0.0486 m^2^ is the wing surface, and b=540 mm is the wingspan of the UAV. The results as a function of the angle of attack are displayed in Figure 9.

In the results, we see that at $\mathrm{AoA}=0^{\circ }$, the rolling coefficient is very low, so the grids without an angle ($\alpha =0^{\circ }$) may not be sufficient to start a balancing manoeuvre. However, from a 5° AoA, the grids increase the rolling coefficient to a value above $C_{l}=0.04$ at 5° and 10° and close to or above $C_{R}=0.06$ for values above $\alpha =15^{\circ }$.

The case with a progressive grid angle (GA 5-10-15) significantly improves the aerodynamic ability of the vehicle to generate a balancing force. From the results in Figure 9, it can be observed that its capability to generate a rolling moment $C_{l}$ increases in all cases between −5° and 10°.

Specifically, under an AoA of −5°, conventional grids generate a negative moment, so they cannot be used for turning manoeuvres. However, the GA 5-10-15 generates a positive moment of $C_{l}$ = 0.02. When the UAV is flying under cruise conditions with AoA = 0°, standard grids generate almost no aerodynamic momentum. However, changing the angle of the grids is highly effective, as they generate a significant moment of almost $C_{l}$ = 0.06 for the turn. Finally, when the UAV is in an ascent situation, controllability for the roll can also be accomplished with this grid angle configuration. At 5° and 10° AoAs, the selected angled grid configuration (GA 5-10-15) continues to generate a roll moment around the aforementioned $C_{l}$ = 0.06, which allows the grids to continue to be used in this configuration and deployed in a semi-aerofoil to perform turns and roll control.

### 3.5. Flow Visualization

Boundary values of the pressure coefficient and non-dimensional velocity on the surface of the UAV are presented in Figure 10, Figure 11, Figure 12 and Figure 13. In these figures, they are presented for the four configurations of grid angles analyzed (GA5, GA10, GA15, and GA 5-10-15) and for two different flight conditions ($\alpha =0^{\circ }$ and $\alpha =5^{\circ }$).

The pressure coefficient is obtained as
(2)$$C_{p}=\frac{\left(p-p_{ref}\right)}{q_{ref}}$$
where $p_{ref}$ and $q_{ref}$ are the reference static and dynamic pressures from the domain inlet.

The non-dimensional velocities presented are computed as
(3)$$V^{*}=\frac{V_{magnitude}}{U_{\infty }}=\frac{\sqrt{u^{2}+v^{2}+w^{2}}}{U_{\infty }}$$
where u, v, and *w* are the three components of the velocity, and $U_{\infty }$ is the inlet velocity of 16 m/s set at the inlet boundary.

Figure 10 presents the pressure coefficient and non-dimensional velocity contours for the GA 5 configuration. The pressure coefficient shows that at AoA = 0° there is a suction peak (negative $C_{p}$, with a value close to −1.5) at the leading edge of the first grid (Figure 10A). However, the rest of the grids do not experience such a suction peak. Therefore, it can be said that they do not contribute significantly to the global lift of the vehicle. On the full wing, a pressure drop is also observed on the top surface, at around 25% of the chord (Figure 10B). The velocity contours show that both the grids and the entire wing behave similarly. That is, acceleration zones appear at the beginning of the profiles and zones where the velocity drops to values below 50% of the free stream (Figure 10C).

By increasing AoA to 5°, the suction peak appears again and with greater intensity ($C_{p}\;\sim -3$) only in the first grid (Figure 10D). It should be noted that the effective angle of the grids in this case is 10°, since this is the sum of the AoA of the vehicle and the value of $\alpha =5^{\circ }$. On the wing, the suction peak is also produced, but it is displaced towards the leading edge (Figure 10E). On the other hand, the velocity contours on the surfaces of the grids reveal that there is an interaction between the flow generated by the grids and the UAV wing (Figure 10F), and small areas of detachment appear on the grids (Figure 10G).

The GA 10 case shown in Figure 11 is similar to the previous case. The differences in pressure coefficients appear at the edge of the first grid (Figure 11A), with a larger suction zone due to the larger angle of attack present in this configuration. As the AoA is increased to 5°, the same effect of a displacement of the negative $C_{p}$ zone towards the leading edge of the wing is observed (Figure 11B). The velocity contours on the grids indicate that there is hardly any detachment on the grids when AoA = 0° (Figure 11C), but that increasing the AoA angle to 5° increases the effective angle to 15°, indicating that they are already highly detached (Figure 11D).

The following case presented in Figure 12 shows the results of pressures and velocities on the UAV surfaces with the grids in the GA 15 configuration. This high angle of attack for the grids means that when flying in cruise with an AoA of 0°, all the grids have a significant suction peak at their leading edge (Figure 12A). On the other hand, the velocity contours reveal that the high angles of the grids generate areas of flow detachment on their surfaces (Figure 12B). However, when the condition is changed to slight ascent (AoA = 5°), the effective angle of the grids is very large (20°) and the grids do not support the adverse pressure gradient, resulting in larger flow detachments (blue areas on the grids, Figure 12C) that almost completely disable the extrados of the grids. At the same time, it can also be observed that there is no longer a suction peak on the grids (Figure 12D).

The last case tested corresponds to grids with incremental angle GA 5-10-15. This was the selected configuration for AoA = 0°, from the aerodynamic efficiency point of view. Thus, in the pressure contours, negative suction peaks can be seen in all three grid profiles, indicating that all three grids work effectively (Figure 13A). In addition, the velocity contours show that compared to the previous configuration (GA 15), the airfoils do not present detached regions and low velocities on the top surface, resulting in a higher aerodynamic efficiency in this configuration and AoA (Figure 13B). As the AoA increases to 5°, the peak suction increases, especially in the first of the grids (Figure 13C); the velocity contours also reveal significant regions of shedding that could explain why the grids do not improve the aerodynamic performance at 5°. However, these low velocity regions (Figure 13D) are less extensive than in the previous case of GA 15.

The visualization results of the pressure coefficient values and velocities displayed on the UAV surfaces show that the maximum efficiency results at AoA = 0° with the GA 5-10-15 grid configuration are produced by the effective operation of the three grids and the low level of flow detachment above them under this flight configuration.

Figure 14 shows the pathlines near the vehicle for the optimum configuration: GA 5-10-15. The pathline color indicates the value of the non-dimensional velocity value ($Vmag/U_{\infty }$). In the detailed views of the grid area, three things can be observed: First, the vortex generated at the wing tip is of very small size in both cases of AoA = 0° and 10° (Figure 14A). Second, the streamlines remain attached along the entire surface of the grids, which ensures that they are working and contributing to the global aerodynamics of the vehicle (Figure 14B). Finally, in the case of AoA = 5°, small interaction areas are observed between the end of the main wing and the grids; these should be analyzed in more detail in future aerodynamic studies with this configuration (Figure 14C).

## 4. Conclusions

In this paper, a numerical analysis of a biomimetic unmanned aerial vehicle (UAV) with three grids at the tip—to simulate the primary feathers of birds—was presented. The study was centered on the aerodynamic effects of changing the dihedral and angle of attack of the grids with respect of the rest of the wing. The aerodynamic performance (lift, drag, and efficiency), rolling capability, and pressure and velocity on the UAV surfaces were obtained under different flight conditions. Since previous studies determined that the improvement in flight efficiency using the grids occurred only in this small range of angles of attack, the simulations with different dihedrals and angles were analyzed in the range of AoAs between −5° and 10°, which is also the typical operating range of an UAV.

In general, the lift and drag coefficients of dihedral grids show very similar values to the case with unmodified grids, and they did not generate an advantageous aerodynamic effect in the UAV studied. The change in the angle of grids increased the $C_{L}$ in a significant way when the UAV was flying at −5° and 0°, but with less effect when the UAV was in ascent flight at 5° or 10°. Similar to lift, the drag coefficient increased in all cases of grid angles tested and was higher as the angle of the grids (α) increased. In addition, the aerodynamic efficiency at low angles of attack (AoA = 0°) increased by 1.4 points for the GA5 configuration, by 2.1 points for the GA10 and GA15 cases, and by up to 2.6 points (the optimal results) for the progressive GA 5-10-15. However, when the UAV ascends (AoA = 5°), changing the angle of the grids no longer brings benefits, as this results in a lower aerodynamic efficiency than the base case.

The grid angle with respect to the wing can be used on only one side of the aircraft to provide a rolling moment to the UAV. In this way, a roll coefficient ($C_{l}=0.06$) can be generated by the grids during manoeuvres at 5° and 10° AoAs using the selected angle grid configuration (GA 5-10-15).

Boundary values of the pressure coefficient and non-dimensional velocity on the surface of the UAV were also presented. The GA 5 configuration presented pressure coefficients with a suction peak at the leading edge of the first grid, whose intensity increased when the AoA increased to 5°. The velocity contours on the surfaces of the grids revealed small areas of flow detachment and an interaction between the flow generated by the grids and the UAV wing. The GA 10 case showed a larger suction area above the grids at AoA = 0°. For GA 10 at AoA = 5° and for GA15 at AoA = 0° and 5°, the effective angle of the grids was too high and the grids did not support the adverse pressure gradient, resulting in larger flow detachments that almost completely disabled the working of the grids. Finally, the incremental angle grid GA 5-10-15 showed the best results, with negative suction peaks in all leading edges. In addition, the small airfoils did not present detached regions at AoA = 0° or smaller detached regions at 5°, resulting in a higher aerodynamic efficiency of the UAV in this configuration.

In summary, increasing the angle of attack of the grids with respect to the wing has very significant benefits in cruise flight, increasing the aerodynamic efficiency by up to 2.6 points; this can be translated into a longer range and endurance and a lower energy consumption during the UAV’s life cycle.

## Figures and Tables

**Figure 1 biomimetics-09-00012-f001:**
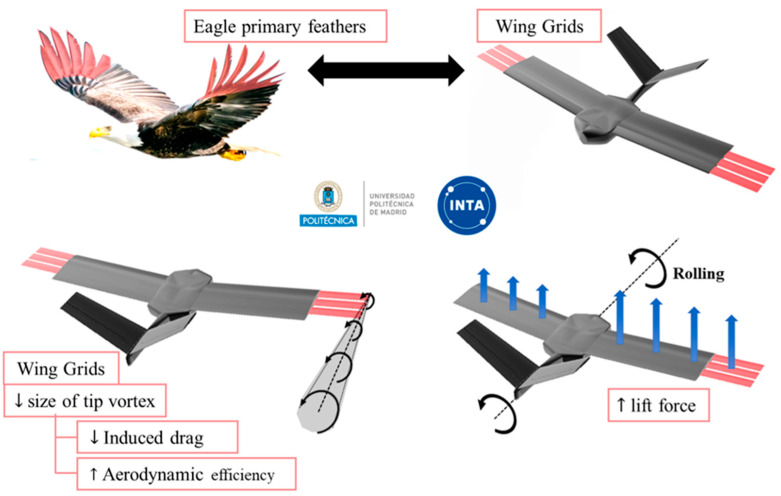
Biomimetic UAV wing grid inspiration on primary feathers of an eagle. Effect of the grids in the reduction of tip vortex size and rolling control.

**Figure 2 biomimetics-09-00012-f002:**
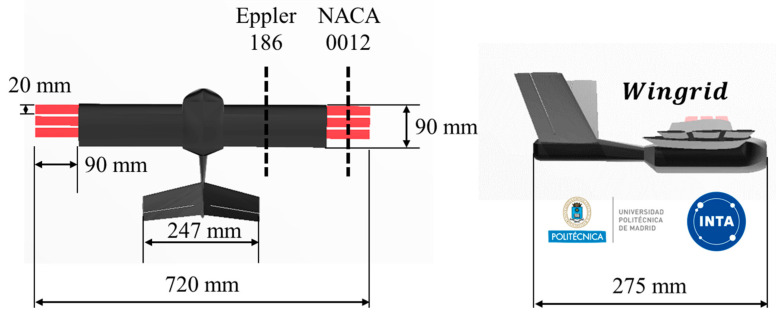
Biomimetic UAV wing grid dimensions with full extended grids at the wingtips.

**Figure 3 biomimetics-09-00012-f003:**
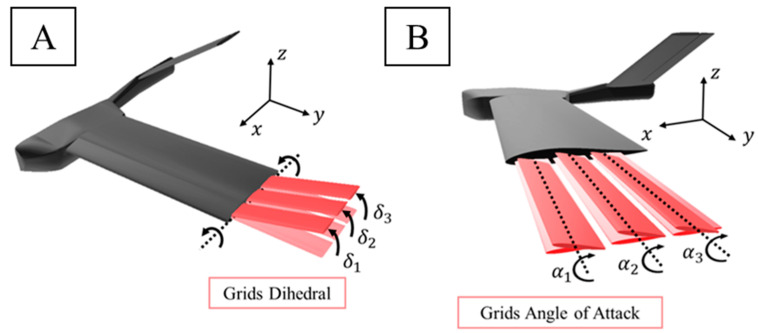
(**A**) Dihedral angle of grids and (**B**) angle of attack of the grids.

**Figure 4 biomimetics-09-00012-f004:**
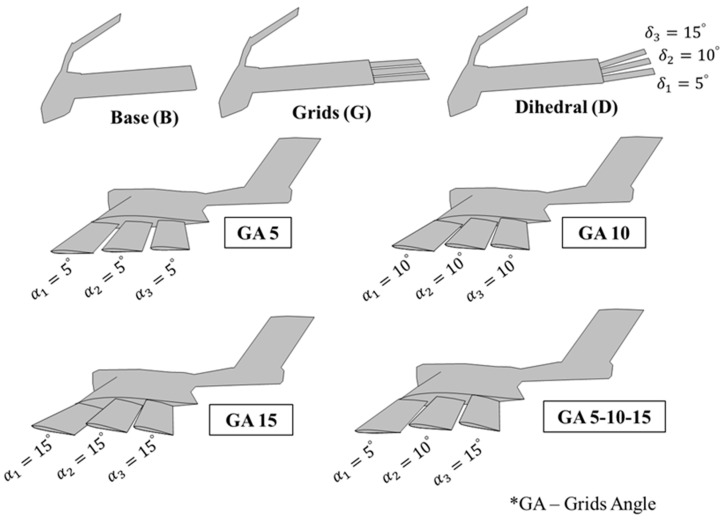
Base configuration (no grids), grids extended, dihedral, and grid angle configurations.

**Figure 5 biomimetics-09-00012-f005:**
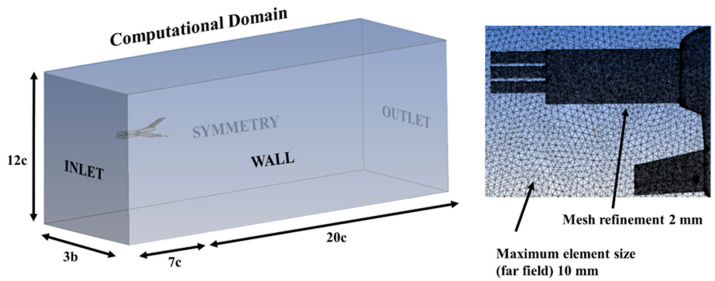
Computational domain size and boundary section names; details of the unstructured mesh for wing grid CFD simulations.

**Figure 6 biomimetics-09-00012-f006:**
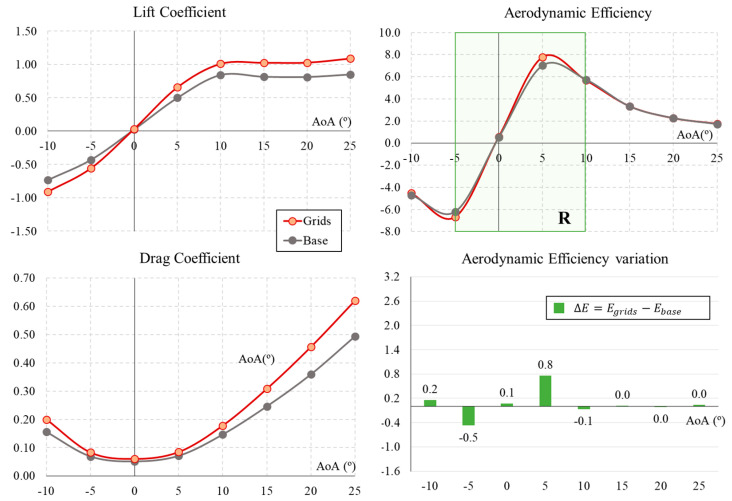
Aerodynamic coefficients and efficiency for the UAV base and grid configurations.

**Figure 7 biomimetics-09-00012-f007:**
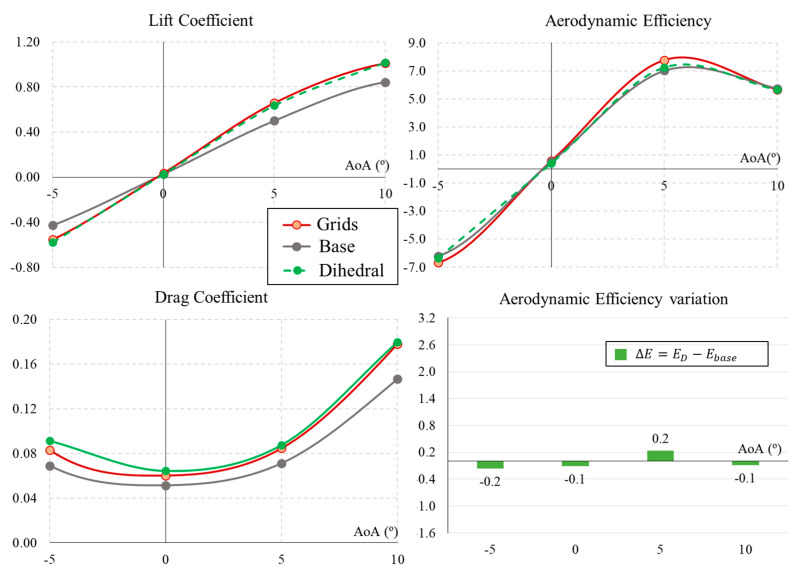
Aerodynamic coefficients and efficiency for the UAV with dihedral grids (D).

**Figure 8 biomimetics-09-00012-f008:**
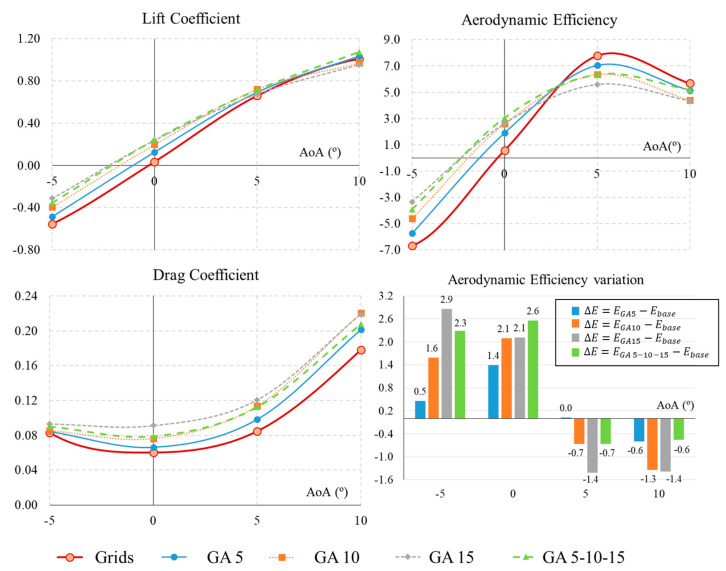
Aerodynamic coefficients ($C_{L},C_{D})$ and efficiency (E) for the UAV grids and AoA grid angle configurations (GA).

**Figure 9 biomimetics-09-00012-f009:**
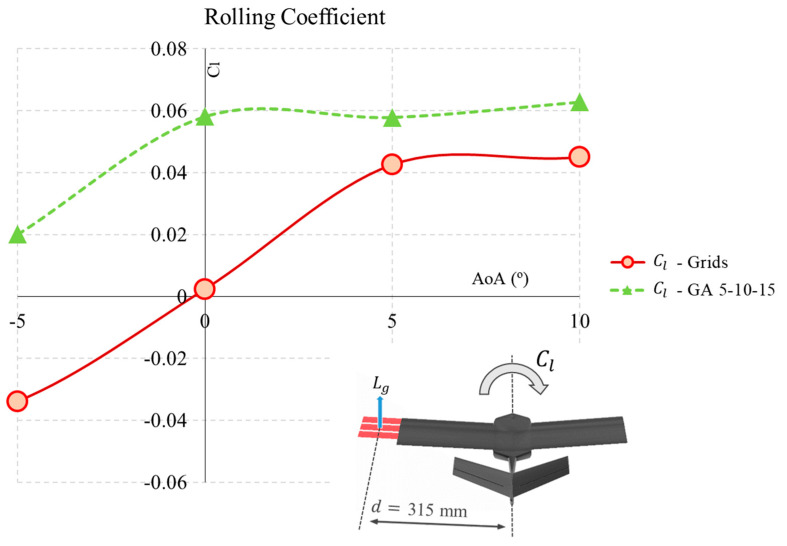
Rolling moment coefficient using grids.

**Figure 10 biomimetics-09-00012-f010:**
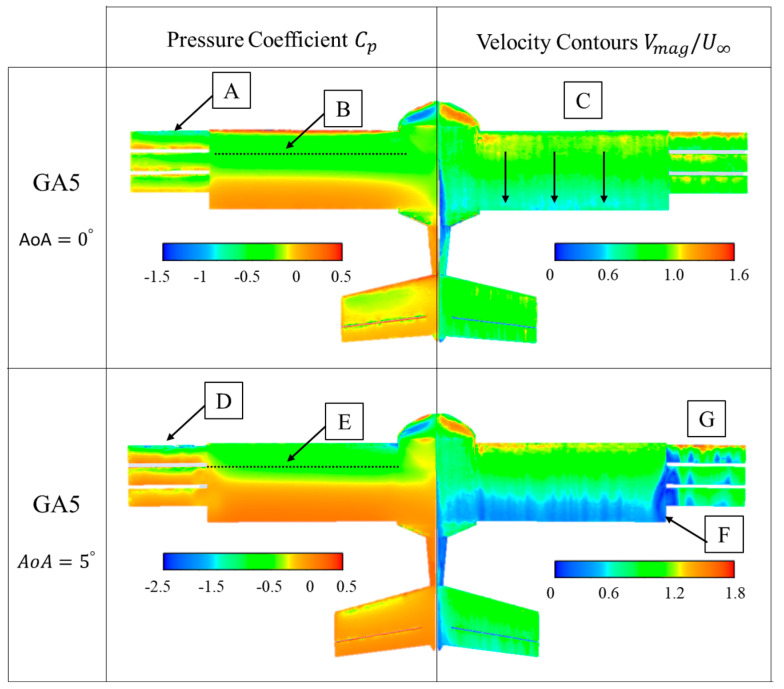
GA 5 boundary values of pressure coefficient ($C_{p}$) and non-dimensional velocities ($V^{*}$).

**Figure 11 biomimetics-09-00012-f011:**
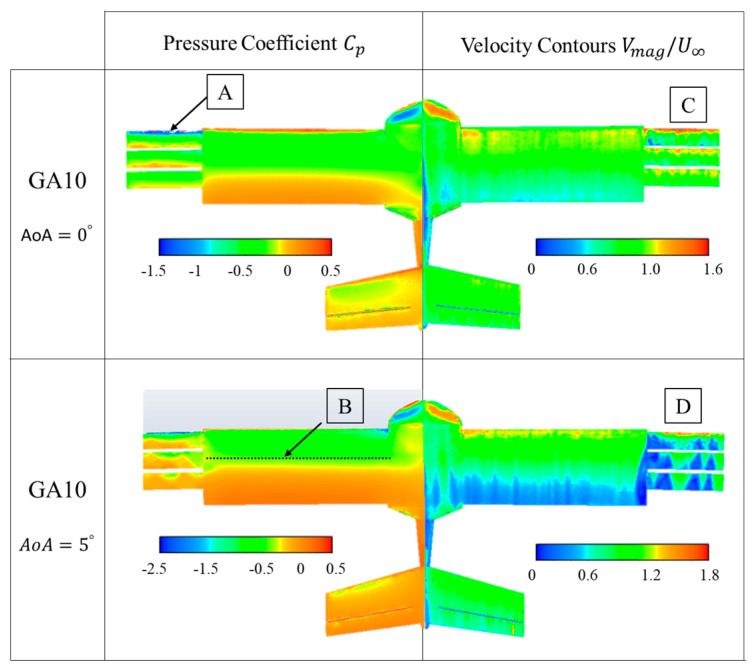
GA 10 boundary values of pressure coefficient ($C_{p}$) and non-dimensional velocities ($V^{*}$).

**Figure 12 biomimetics-09-00012-f012:**
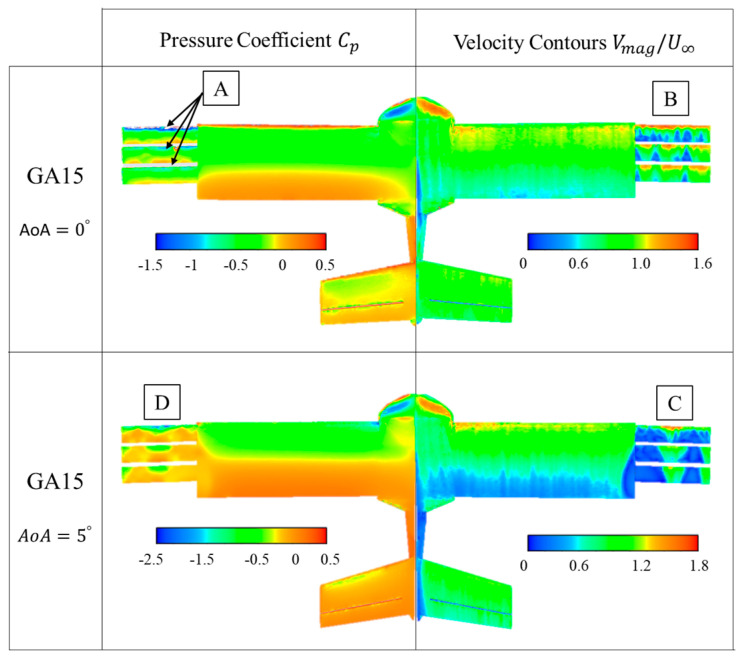
GA 15 boundary values of pressure coefficient ($C_{p}$) and non-dimensional velocities ($V^{*}$).

**Figure 13 biomimetics-09-00012-f013:**
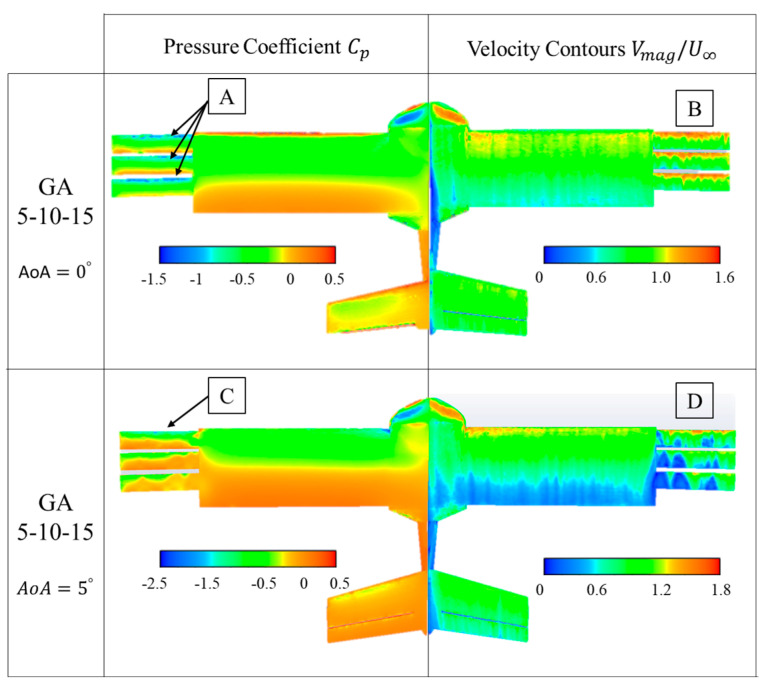
GA 5-10-15 boundary values of pressure coefficient ($C_{p}$) and non-dimensional velocities ($V^{*}$).

**Figure 14 biomimetics-09-00012-f014:**
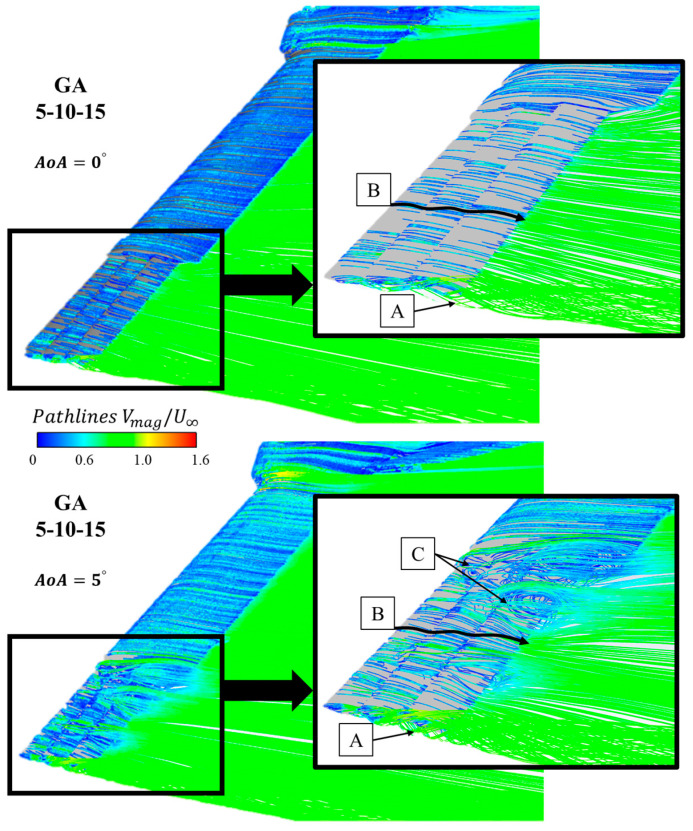
GA 5-10-15 pathlines of non-dimensional velocity around the grids for AoA = 0° and 5°. (**A**) Tip vortex, (**B**) Streamlines attached, (**C**) Interaction area between grids and wing.

## Data Availability

Data are available on request due to restrictions (e.g., privacy or ethical). The data presented in this study are available on request from the corresponding author.

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
