# Peer review of "Computational Study of Aerodynamic Effects of the Dihedral and Angle of Attack of Biomimetic Grids Installed on a Mini UAV"

_biomimetics, 2023, doi:10.3390/biomimetics9010012_

Round 1

Reviewer 1 Report

Comments and Suggestions for Authors

In a very interesting manuscript entitled Computational study of aerodynamic effects of the dihedral and angle of attack of biomimetic grids installed on a mini UAV”, the authors present a numerical analysis of a biomimetic UAV with three grids at the tip that simulates primary feathers of birds. I think this paper can be very useful in terms of UAV aerodynamic design optimization but it should be revised taking into account some minor issues.

As a general comment, this manuscript seems to be interesting and original, and the submission has the appropriate length to be understood. Furthermore, the manuscript is well constructed, the title and abstract are appropriate for the content of the text and the methodology that the authors developed is well discussed. The selected references are prevailingly well related to the theme.

In Flight Dynamics of an unmanned aerial vehicle (UAV), we usually use the designations yaw, pitch, and roll/bank for angles. I am wondering if I am interpreting equation (1) correctly because de symbols used are not usual in the nomenclature of Flight Dynamics. We should try to cite primary sources whenever possible; in this case, can the authors reference (and make the in-text citation) the book they got the formula (1)?

Nowadays abbreviations are used in aeronautics for terms, definitions, standards, etc. The expression “Angle(s) of attack” appears 16 + 5 = 21 times in the manuscript which justifies the use of the abbreviation: AoA (angle of attack), along the text.

There are important positive features in the manuscript; for instance, there are some interesting figures of good quality. The results obtained demonstrate the novelty and originality of the numerical analysis of a biomimetic UAV presented. The authors did a good job.

Reviewer 2 Report

Comments and Suggestions for Authors

Comments on the Quality of English Language

The English is mostly fine, but it is difficult to read in spots.  It requires editing before moving forward.

Reviewer 3 Report

Comments and Suggestions for Authors

This paper deals with modeling additional components on the wingtip of a mini UAV. The CFD tool - ANSYS Fluent was used for the analysis. I think this paper looks pretty good, but it's missing a few things.

1/ In my opinion, the described methodology is the worst. These authors wrote "Near the aircraft body, sizing restrictions were imposed, dictating a maximum element size of 2 mm". What is more important for the reader is what the values of the wall y+ parameter are, especially since "standard wall functions were implemented". Nothing has been written about solver settings. This needs to be improved. Nothing was mentioned about the calculation itself. The grid was described rather poorly. In Fig. 5 you can see the mesh, but you can't see what the mesh looks like next to the wall. An additional photo is needed. There is an additional photo of the domain in the same drawing. These dimensions need to be better shown. It is not clear what the dimensions of the UAV itself are, and dimension 7c is not clearly shown.

2/ This paper lacks reference to the experiment or research of other authors. The CFD model is not the best choice. I mean the turbulence model. Therefore, large errors can be expected. I understand that an experimental model is not always available. However, you can examine, for example, the airfoils themselves. For these Reynolds numbers, this is not a trivial task.

3/ it would be worth enriching the results in this article. An additional subsection showing the visualization of the flow around the tips for different cases was an asset.

Round 2

Reviewer 2 Report

Comments and Suggestions for Authors

Thank you for addressing my comments.  The only remaining issue I have is with the Figures (as mentioned previously).  The change of the colormap scales highlights the difficulties of using rainbow colormaps.  Looking at Figure 11 (which corresponds to Figure 2 in the author response to me) as an example: the trailing edge of the velocity contours looks radically different.  I feel like this changes the results enough that it still needs attention.

Author Response

The trailing edge of velocity contours indeed looks radically different. But the velocity values presented in both are the same (around 1.2 values on the leading edge, 0.8 on the middle part of the upper profile, and 0.6 on the trailing edge). It is because the scale range of the legend has been reduced and the changes in velocities above the wing surfaces are now easier to observe, with a wider range of color for the same values.

Contrast has been increased to Figures 10, 11, 12, and 13 to improve the visualization of the color contours. 

Reviewer 3 Report

Comments and Suggestions for Authors

This manuscript has been revised well enough. I recommend it for publication in this form.

Author Response

Thanks for your review.